# How to Improve the Reactive Strength Index among Male Athletes? A Systematic Review with Meta-Analysis

**DOI:** 10.3390/healthcare10040593

**Published:** 2022-03-22

**Authors:** André Rebelo, João R. Pereira, Diogo V. Martinho, João P. Duarte, Manuel J. Coelho-e-Silva, João Valente-dos-Santos

**Affiliations:** 1LX Applied Sport Sciences Research Group, 1000-289 Lisbon, Portugal; andre94rebelo@hotmail.com (A.R.); dvmartinho92@hotmail.com (D.V.M.); joaopedromarquesduarte@gmail.com (J.P.D.); j.valente-dos-santos@hotmail.com (J.V.-d.-S.); 2CIDEFES, Centro de Investigação em Desporto, Educação Física e Exercício e Saúde, Universidade Lusófona, 1749-024 Lisboa, Portugal; 3COD, Center of Sports Optimization, Sporting Clube de Portugal, 1600-464 Lisbon, Portugal; 4Research Unity in Sport and Physical Activity (CIDAF, UID/DTP/04213/2020), Faculty of Sport Sciences and Physical Education, University of Coimbra, 3040-248 Coimbra, Portugal; mjcesilva@hotmail.com; 5Polytechnic of Coimbra, Coimbra Health School, Dietetics and Nutrition, 3045-093 Coimbra, Portugal; 6Laboratory for Applied Health Research (LabinSaúde), 3045-093 Coimbra, Portugal; 7Porto Biomechanics Laboratory (LABIOMEP-UP), University of Porto, 4200-450 Porto, Portugal

**Keywords:** strength, power, reactive strength, players, plyometric training, resistance training

## Abstract

The reactive strength index (RSI) describes the individual’s capability to quickly change from an eccentric muscular contraction to a concentric one and can be used to monitor, assess, and reduce the risk of athlete’s injury. The purpose of this review is to compare the effectiveness of different training programs on RSI. Electronic searches were conducted in MEDLINE, PubMed, Scopus, SPORTDiscus, and Web of Science from database inception to 11 February 2022. This meta-analysis was conducted in accordance with the recommendations of the preferred reporting items for systematic reviews and meta-analyses (PRISMA). The search returned 5890 records, in which 39 studies were included in the systematic review and 30 studies were included in the meta-analysis. Results from the randomized studies with the control group revealed that plyometric training improved RSI in adult athletes (0.84, 95% CI 0.37 to 1.32) and youth athletes (0.30, 95% CI 0.13 to 0.47). Evidence withdrawn from randomized studies without a control group revealed that resistance training also improved the RSI (0.44, 95% CI 0.08 to 0.79) in youth athletes but not in adults. Interventions with plyometric training routines have a relatively large, statistically significant overall effect in both adult and youth athletes. This supports the implementation of this type of interventions in early ages to better cope with the physical demands of the various sports. The impact of resistance training is very low in adult athletes, as these should seek to have a more power-type training to see improvements on the RSI. More interventions with sprint and combined training are needed.

## 1. Introduction

The reactive strength index (RSI) describes the individual’s capability to quickly change from an eccentric muscular contraction to a concentric one [1]. In other words, the RSI was created to assess the athlete’s reactive strength, and it was originally measured with the drop jump (DJ) test [1]. For this test, the athletes must perform a vertical jump as soon as they land on the ground from a specific height [1]. The hands can stay on the athletes’ hips throughout the test or not, as both methods have shown good levels of reliability [2]. This test should incorporate various drop heights to assess at what height the athlete can elevate more his centre of gravity, and it was already proven to be a reliable and valid test to measure the RSI [3]. The RSI can be calculated by dividing the jump height by the ground contact time, providing valuable information for coaches, regarding plyometric performance (i.e., jump height) and how each jump is performed (i.e., ground contact time) [4]. Jump height can be measured directly or can be derived from flight time with the following mathematical formula [5]: jump height (m) = (gravity × (flight time)^2^)/8, where gravity = 9.81 m/s and flight time is in seconds.

Most recently, with advancements in technology, more tests have been developed to measure the RSI, such as the countermovement (CMJ), tuck jump, squat jump, weighted CMJ, single-leg jump [6], 10/5 [7], single rebound jump [8], vertical rebound for 5 repetitions [9], vertical rebound for 15 repetitions [10], and vertical rebound for 10 s tests [11]. In those cases where there is no drop or rebound jump, the RSI is designated explicitly by reactive strength index modified (RSI_mod_), since it is calculated by dividing the jump height by the time to take-off (time to produce force from the beginning of the eccentric muscular phase until the moment the athlete leaves the ground) [6].

To obtain these variables mentioned before (i.e., jump height, flight time, ground contact time, and time to take-off), three different methods can be used: (a) the flight time [12]; (b) the difference between the height of two marks during the jump [13]; and (c) the mathematical integration of the ground reaction force [14]. The first one requires the use of contact mats [12,15,16], photocell mats [13,15], or accelerometers [16,17]. The second method uses different devices to calculate displacement (i.e., linear position transducers) [18]. The third method is considered the best one, as its accuracy is extremely high if adequate sampling frequency methods are chosen and requires the use of one or two force plates [13,19].

The use of RSI is vital for high-performance sports professionals, as it can be used as a motivational tool, in a way that coaches can instantly deliver feedback to their athletes, according to their RSI value, in order to improve their physical performance [5]. Furthermore, both RSI and RSI_mod_ can be used as variables to potentially monitor athlete’s neuromuscular readiness [20]. Moreover, the RSI has been shown to have a strong relationship with change of direction speed, acceleration speed [21], and agility [22]. Additionally, maximal strength, especially relative to body mass, appears to have a very strong relationship with RSI_mod_, indicating that stronger athletes tend to have better reactive strength [23].

However, there is no clear information on which type of training would produce better improvements on RSI. Besides that, to the best of our knowledge, the scientific literature does not review this topic. Therefore, the aim of this study was to analyse the strategies that can improve the RSI of male athletes, through a systematic review of experimental research and meta-analysis.

## 2. Methods

### 2.1. Protocol and Registration

This meta-analysis was conducted in accordance with the recommendations of the preferred reporting items for systematic reviews and meta-analyses (PRISMA) [24]. The study protocol was registered with PROSPERO (CRD42020176616).

### 2.2. Information Sources

The literature search on five electronic databases (i.e., MEDLINE, PubMed, Scopus, SPORTDiscus, and Web of Science) started on 5 July 2020 and was conducted from database inception to 11 February 2022.

### 2.3. Search Strategy

All retrieved papers were exported to CADIMA software, a tool designed to increase the efficiency of the evidence synthesis process and facilitate reporting of all activities to maximize methodological rigor [25]. Duplicates were automatically removed. Titles and abstracts of potentially relevant papers were screened by two reviewers (A.R. and J.R.P.). Disagreements between authors were solved through discussion and, when necessary, three other authors (D.M., J.P.D., and J.V.-d.-S.) were involved. Full-text copies were acquired for all papers that met title and abstract screening criteria. Full-text screening was performed by two reviewers (A.R. and J.R.P.). Again, any discrepancies were discussed, until the authors reached an agreement and consulted the three other authors, when required.

The comprehensive search strategy is available in the Appendix A.

### 2.4. Inclusion Criteria

Scientific peer-reviewed published papers written in English, Portuguese, French, and Spanish were eligible for the present systematic review. The review sought to identify all studies reporting exercise interventions to improve the RSI in both male adult and youth athletes. Therefore, studies were eligible if: (1) subjects were male athletes; (2) subjects had between 11 and 45 years old; (3) the study included at least two moments of evaluation, with a baseline RSI measurement and post-intervention RSI measurement; (4) the study included a training program that aimed to improve the RSI.

### 2.5. Exclusion Criteria

Studies that do not describe a protocol to induce effects on RSI or that used RSI during a recovery program were excluded from the present study.

### 2.6. Categorisation of Studies

We identified six categories of exercise interventions, through the process of reviewing the included studies. The definitions of these exercise interventions are provided in Table 1.

### 2.7. Data Extraction

Pre-established data extraction criteria were created with seven items: (a) general information (authors name and year of the study), (b) sample characteristics (size and age), (c) sport, (d) training program, (e) measurement equipment, (f) methodology, and (g) results.

### 2.8. Risk of Bias Assessment

The revised Cochrane risk of bias tool for randomized trials (RoB 2.0) scale was used to quantify the risk of bias in eligible, individually randomized, parallel-group trials and provide information on the general methodological quality of studies. The RoB 2.0 scale rates internal study validity and the presence of replicable statistical information on a scale from low risk of bias to high risk of bias [26]. The risk of bias in non-randomized studies of interventions (ROBINS-I) tool was used to quantify the risk of bias in eligible non-randomized trials and provide information on the general methodological quality of the studies. The ROBINS-I scale rates internal study validity and the presence of replicable statistical information on a scale from 1 (low risk of bias) to 4 (high risk of bias) [27]. Inter-rater agreement was calculated using Cohen’s kappa coefficient (k). Using different tools to assess the risk of bias on randomized and non-randomized studies was supported elsewhere [28].

### 2.9. Statistical Models

Meta-analyses were conducted to estimate the overall effects of the intervention programs to improve RSI for all available data, as well as for studies that only included randomized samples. Within the previous categorization, meta-analyses were also divided by studies with (intervention vs. control) and without a control group (pre-intervention vs. post-intervention). Additionally, subgroup meta-analyses were performed to identify the effects of each training program (“plyometrics”, “resistance training”, “sprint training”, “sprint or plyometric or a combination of those”, and “sports-specific training”) on adults (≥18 years old) and youth (<18 years old) athletes. Additionally, pre- and post-intervention results from all the included studies (randomized and non-randomized) were ranked on two different meta-analyses to understand the effectiveness of each training program on adult and youth athletes.

Meta-analysis, and the respective forest plots, were calculated when the mean, standard deviation, and sample sizes were introduced on the Cochrane collaboration’s review manager computer program (RevMan version 5.4.1, Oxford, UK). When this data was impossible to retrieve from the manuscript [29,30,31,32,33,34,35,36,37,38], authors were contacted to provide the missing information. Most of the authors replied to the request [29,30,31,32,33,34]; therefore, the data was included on the meta-analyses. In addition, when the same study reported different RSI measurements (i.e., from different heights and/or tests), the method that resulted in the highest positive performance change was selected for the meta-analysis.

The pooled data for each outcome were reported as standardized mean differences (SMD), with a 95% confidence interval (CI). Each meta-analysis was performed using the random-effects model, and heterogeneity was assessed using I^2^ statistic and chi-square (Q) tests. A Q value with a significance of *p* ≤ 0.05 was considered significant heterogeneity, while, for the I^2^ value, 25% was considered low, 50% was considered moderate, and 75% was considered high heterogeneity [39].

## 3. Results

### 3.1. Study Selection

The search strategy returned 5890 records, and the PRISMA flow diagram [40] is shown in Figure 1. Records were excluded based on the included participants (not male athletes), intervention or comparator (not a training program), post-intervention data (not reporting RSI measurements), or testing on surfaces other than the floor (i.e., force sledge). In addition, for the quantitative synthesis (meta-analysis), ineligible studies were excluded for reporting RSI data only in figures and/or percentage [35,36,37,38] or for being unique, in terms of physical training method [41,42] and, therefore, not being able to pair it with other studies to conduct a meta-analysis. In total, thirty-nine studies were included in the systematic review and thirty-three were included in the meta-analysis.

### 3.2. Risk of Bias Assessment

Inter-rater agreement for the risk of bias assessment, using the RoB 2.0, was κ = 0.933; for the ROBINS-I, it was κ = 1.0. Thus, overall, the risk of bias within individual studies assessed using the RoB 2.0 scale ranged between low and high risk of bias (Appendix A), whereas the ROBINS-I scale ranged from a serious to critical risk of bias (Appendix A).

Of the thirty-three randomized studies assessed with the RoB 2.0, twelve (36%) [35,43,44,45,46,47,48,49,50,51,52,53] had a high risk of bias. Randomisation process (18%) and selection of reported results (15%) were the most common sources of high risk of bias.

Of the six non-randomized studies assessed with the ROBINS-I, one (17%) [54] was of critical risk of bias assessment, due to their departures from the intended interventions, participants being excluded because of missing data, and selection of participants based on their characteristics observed after the start of the intervention. The remaining five studies (83%) [36,42,55,56,57] were of serious risk of bias assessment.

### 3.3. Study Characteristics

Study characteristics (including training program characteristics) for all 39 included studies are presented in Appendix A. Overall, the most common tests used to quantify the RSI are DJs (79%) [29,30,31,32,33,34,35,36,38,41,42,43,45,46,49,50,51,52,53,54,56,57,58,59,60,61,62,63,64,65,66], vertical hops (21%) [44,47,48,67,68,69,70,71], and CMJs (8%) [37,55,72]. The most popular materials used to quantify the RSI are contact mats (46%) [29,30,31,32,33,34,36,38,45,46,49,53,61,62,68,69,71,72], force plates (36%) [35,37,41,42,43,48,50,52,54,55,56,60,65,66], and photoelectric systems (15%) [44,47,57,58,67,70]. Soccer is the most common sport studied (56%) [29,30,31,32,33,34,36,38,44,45,46,47,48,50,52,53,61,62,67,68,69,70], followed by rugby (18%) [35,43,50,52,58,60,65] and basketball (8%) [50,53,72]. Intervention duration ranged from four weeks [35,42,65] to two years [36].

### 3.4. Meta-Analysis

#### 3.4.1. Studies with Control Group

Fourteen randomized studies (four with adult athletes [50,51,53,60] and ten with youth athletes [29,30,31,32,33,34,45,61,62,68]), with the control group, reporting plyometric training programs were included in the meta-analysis (Figure 2a). The overall standardized mean difference in adult athletes was 0.84 (95% CI 0.37 to 1.32), with a significant overall intervention effect. The overall standardized mean difference in youth athletes between groups was lower than adult athletes, at 0.30 (95% CI 0.13 to 0.47), with the intervention effect almost the same. Examination of heterogeneity statistics revealed a non-significant heterogeneity for both adult and youth athletes RSI results (χ^2^ = 13.60 (*p* = 0.06), I^2^ = 49% and χ^2^ = 23.86 (*p* = 0.16), I^2^ = 25%, respectively).

The meta-analysis included two randomized studies (one with adult athletes [52] and another with youth athletes [68]) reporting a combination of change of direction, plyometric, and/or sprint training programs (Figure 2b). The overall standardized mean difference in adult athletes was −0.01 (95% CI −0.83 to 0.82), with a non-significant overall intervention effect. The overall standardized mean difference in youth athletes between groups was higher than adult athletes, at 0.65 (95% CI −0.02 to 1.33), with a non-significant overall intervention effect.

Regarding the non-randomized studies, in Appendix A, three non-randomized studies with adult athletes [43,54,56] and one non-randomized studies with youth athletes [55] reporting resistance training programs can be seen. The overall standardized mean difference in adult athletes was 0.49 (95% CI 0.02 to 0.96), with a significant overall intervention effect. The overall standardized mean difference in youth athletes between groups was higher than adult athletes, at 0.65 (95% CI 0.15 to 1.16), also with a significant overall intervention effect. Examination of heterogeneity statistics revealed a non-significant heterogeneity for both adult and youth athletes RSI results (χ^2^ = 0.49 (*p* = 0.92), I^2^ = 0% and χ^2^ = 0.12 (*p* = 0.94), I^2^ = 0%, respectively). Non-randomized studies reporting plyometric interventions or a combination of change of direction, plyometric, and/or sprint training programs were not found. Therefore, no meta-analyses were performed for these interventions with non-randomized studies.

#### 3.4.2. Studies without Control Group

Nine randomized studies (two with adult athletes [49,66] and seven with youth athletes [44,46,47,48,67,69,70]), without a control group, reporting plyometric training programs were included in the meta-analysis (Figure 3a). The studies with adults reported a significant intervention effect, with a standardized mean difference of 0.94 (95% CI 0.29 to 1.60). The overall standardized mean difference in youth athletes was 0.78 (95% CI 0.31 to 1.24), with a significant overall intervention effect. Examination of heterogeneity statistics revealed a high heterogeneity for youth athletes RSI results (χ^2^ = 58.49 (*p* < 0.00001), I^2^ = 76%).

Only one randomized study with adults [49] examined the impact of sprint training methods in RSI (Figure 3b). The study’s standardized mean difference was 0.56 (95% CI −0.11 to 1.23) with a non-significant intervention effect. No studies were found for youth athletes.

Two randomized studies with adult athletes [49,72] and three randomized studies with youth athletes [58,65,72] reporting resistance training programs were included in the meta-analysis (Figure 3c). The overall standardized mean difference in adult athletes was 0.19 (95% CI −0.33 to 0.72), with a non-significant overall intervention effect. The overall standardized mean difference in youth athletes between groups was higher than adult athletes, at 0.44 (95% CI 0.12 to 0.75), with a significant overall intervention effect. Examination of heterogeneity statistics revealed a non-significant heterogeneity for both adult and youth athletes RSI results (χ^2^ = 0.42 (*p* = 0.81), I^2^ = 0% and χ^2^ = 1.54 (*p* = 0.98), I^2^ = 0%, respectively). One additional non-randomized study [57] reporting resistance training was found in youth athletes. However, the mentioned study did not change the significant effect previously calculated only for the randomized studies (Appendix A).

#### 3.4.3. Studies Ranked by Intervention Effects

A ranked forest plot of the intervention effects on adult athletes demonstrates that, compared with pre-test, several interventions demonstrated similar differences: combined training, with an overall standardized mean difference of 0.58 (95% CI −0.06 to 1.21); sprint training, with an overall standardized mean difference of 0.56 (95% CI −0.11 to 1.23); and plyometric training, with an overall standardized mean difference of 0.54 (95% CI 0.26 to 0.82) (Figure 4a). Of those interventions, only plyometrics were considered significant. In youth athletes, combined training had the largest standardized mean difference (1.15, 95% CI 0.48 to 1.83), followed by plyometric training (0.61, 95% CI 0.42 to 0.80) (Figure 4b).

### 3.5. Qualitative Analysis

As previously reported, six studies were not included in the quantitative analysis. Two [37,38] had plyometric training interventions and, like those who were part of the meta-analysis, it was seen that this training method improves the RSI. Like the results from the meta-analyses on plyometric training, participants of those two studies also improved their RSI. Additionally, another study [41] with a plyometric training method inside water also improved the RSI performance of their participants. With respect to resistance training, one study [35] reported two different resistance training interventions in adult athletes, and the findings suggested that some individuals had performance decrements over the four week training period. However, one study [36] with youth athletes showed RSI improvements with a resistance training intervention. Lastly, one study [42], with adult athletes, had a badminton specific/change of direction intervention, and it was seen that the RSI improved, mainly due to ground contact times enhancements, rather than jumping height. The characteristics of all these mentioned studies can be seen in Appendix A.

## 4. Discussion

This is the first study using meta-analyses to investigate the comparative effectiveness of different exercise interventions to improve the RSI in both male adult and youth athletes. Given the importance that RSI may have in athletes’ performance, in particular, acceleration, change of direction, [21], and agility [22], it is vital to investigate effective strategies to improve it. Randomized studies with a control group reported significant overall intervention effects, regarding plyometric and resistance training programs, whereas combined training interventions did not have a significant overall effect. Randomized studies without a control group reported similar tendencies in youth athletes; however, plyometric, sprint, and resistance training programs showed a non-significant overall intervention effect in adult athletes. Compared to randomized studies alone, results from a combination of randomized and non-randomized did not differ. The results ranked by treatment effect indicate that resistance training is inferior, compared to both sprint and plyometric training methods, in enhancing the RSI in adult and youth athletes.

Plyometric training is characterized by a pre-activation (stretch) of the extensors’ muscles (e.g., quadriceps during a jump), followed by a shortening phase of these same extensors’ muscles, which represents the stretch-shortening cycle (SSC) [73]. The duration of the SSC, usually measured by the ground contact time, can be categorized into slow (>250 milliseconds; CMJ, changes of direction) or fast (<250 milliseconds; DJ, sprints) [74]. Studies have shown that low correlations exist between these two types of SSC [75,76,77], and reactive strength training is commonly referred as “plyometrics”.

The effectiveness of plyometric training methods to improve jumping height ability have been shown [75]; since the RSI can be enhanced by improving this variable (by the formula: RSI = jump height/ground contact time), it is not surprising that plyometrics were successful in increasing the RSI in both adult and youth athletes. Nevertheless, one of the plyometric studies [70] reported that the intervention group declined in the RSI after the training intervention. However, it should be noted that this intervention group performed the plyometric training on unstable surfaces, which may be done with longer ground contact times, resulting in worse RSI values.

According to a motor learning perspective, exercises performed in a similar way to the target task produce an enhanced performance, since they generate a greater transfer, due to their specificity [78]. With this under consideration, both plyometric and sprint exercises are usually performed with a maximum acceleration during the triple-extension phase, and the same applies to the various tests used to quantify the RSI (i.e., vertical hops, CMJ, and DJ) [79]. Although some resistance training exercises might also have this triple-extension phase, the movement will always be slower because of the higher loads used, compared to plyometric and sprint training methods [80]. Additionally, sprint training methods enhance the SSC muscle function, by decreasing ground contact time and increasing flight time [81], because of the importance of the eccentric phase during sprinting to maximize the power output during the concentric phase [82]. Moreover, during the maximal velocity phase of a sprint, the ground contact time is minimal (~80–120 ms); therefore, there is no time to produce muscle force [83]. Thus, the force is only produced by the tendons, being that the tendon stiffness is a particularly important property to generate high forces in a truly short time (fast SSC) [84]. The ability to sprint and generate high forces in short periods of time is somehow related to the RSI, as it is calculated by the jump height, divided by the contact time. Consequently, sprints might reduce the denominator of the RSI formula (i.e., contact time), and subsequently, increase the RSI value. Furthermore, during the acceleration phase of a sprint, an athlete needs to produce more horizontal force to propel himself, while, on the maximal velocity phase, an athlete needs to produce more vertical force [83]. Thus, the force vector direction is the same during a vertical jump (used to access the RSI) as it is on the maximal velocity phase of a sprint. Consequently, sprint training (in particular, maximal velocity), might have some transfer to vertical jumps and, therefore, to better RSI performances, since this type of training may also enhance motor unit firing frequency, ultimately benefitting strength−power characteristics [85]. Nevertheless, only few studies that used sprint training to improve the RSI were found and more research is needed to corroborate this tendency.

Although resistance training induced improvements on RSI, youth athletes seem to take more advantage of this type of training, compared to adult athletes. In general, adult athletes are more skilled and have higher levels of strength, compared to youth athletes. Thus, after achieving specific strength standards, to improve their performance, adult athletes must shift towards a power-type training, while maintaining their strength levels [86]. In youth athletes, on the other hand, resistance training enhances motor control and coordination, which are the base for bigger future improvements in other physical qualities, such as velocity and power [86]. These neural changes that youth athletes experience during resistance training might explain why this training method produces better RSI improvements, compared to adult athletes.

The plyometric intervention was the training program most reported in the present review. Nevertheless, due to different designs (i.e., randomized study vs. non-randomized study or presence vs. absence of a control group) on the individual studies, it was not possible to band together with the different plyometric training programs into different categories (e.g., fast SSC, slow SSC, extensive plyometrics, intensive plyometrics, unilateral, bilateral, vertical emphasis, and horizontal emphasis). The resistance training program is also a broad category. Once again, due to different study designs, it was not possible to group these studies into more specific categories (e.g., high volume, strength training, power training, and accentuated eccentric training). As it was previously mentioned, reactive strength describes the individual’s capability to quickly change from an eccentric muscular contraction to a concentric one [1]. Consequently, it is plausible to consider that an athlete that is able to produce higher rates of force development during the eccentric phase will be able to express his/her concentric potential quicker than an athlete with lower levels of eccentric rates of force development. Therefore, it is expected that resistance training focusing on producing more force (eccentrically) in less time will also improve the RSI. Three studies [43,44,65] that were included in the present review reported this type of training. Whereas, in the studies from Murton and colleagues [65] and Douglas and colleagues [43], the adult rugby players in the intervention group improved their performance on RSI after 4 and 12 weeks, respectively, of accentuated eccentric resistance training. In the study from Fiorilli and colleagues [44], the youth soccer players from the intervention group did not improve their RSI after 6 weeks of flywheel eccentric overload training (pre-test: 0.84 ± 0.18; post-test: 0.83 ± 0.18). The flywheel device is an isoinertial equipment that, in a given movement, returns during the eccentric phase, the exact same force produced on the concentric phase. Therefore, using this device does not guarantee that the eccentric overload has been reached per se. As the methodology used by Fiorilli and colleagues [44] was not fully detailed, this may justify the results previously mentioned. Consequently, future research should aim to understand not only the effect of the resistance or plyometric training on RSI but, essentially, the effects of different types/categories of resistance and plyometric training on RSI.

Furthermore, it was noted that only 36% of the studies included used the gold standard method of measurement (i.e., force plates) [13]. Despite the fact that contact mats are cheaper and easier to use than force plates [13], investigators should be warned that the flight times derived from the contact mats are not always consistent, when compared with the flight times derived from the force plates [87]. More than half of the studies included in the review (i.e., 56%) had a sample of soccer athletes. It would be important that future research dedicate itself to study other sport modalities.

This review highlighted the best training intervention to improve the RSI and indicated possible directions for future research in this topic. Nevertheless, a limitation of this review is that some studies could not be included in the meta-analysis, due to not reporting RSI values in a continuous way. Therefore, fewer studies were included in the meta-analysis than in the systematic review; it is possible that the inclusion of these studies could modify the results observed. Additionally, 36% of the randomized studies included had a high risk of bias. Although, it is crucial to notice that most of the high risk of bias was due to the randomisation process.

## 5. Conclusions

Results from this systematic review and meta-analysis suggest that interventions with plyometric training routines have a relatively large, statistically significant overall effect in both adult and youth athletes. Thus, evidence supports implementing these types of interventions at an early age, in order to better cope with the physical demands of the various sports. Resistance training seems to have less impact on trained adult athletes; therefore, trained adult athletes should seek to have a more power-type training to improve the RSI. More research on specific resistance (e.g., strength vs. power type vs. eccentric overload) and plyometric (e.g., fast vs. short SSC) interventions are needed. Likewise, more research with sprint and combined training interventions is needed to understand the effects of these methods on the RSI.

## Figures and Tables

**Figure 1 healthcare-10-00593-f001:**
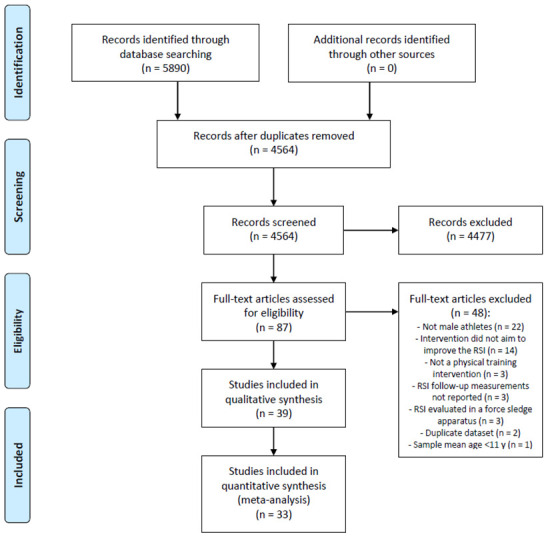
PRISMA statement flow chart. RSI, reactive strength index.

**Figure 2 healthcare-10-00593-f002:**
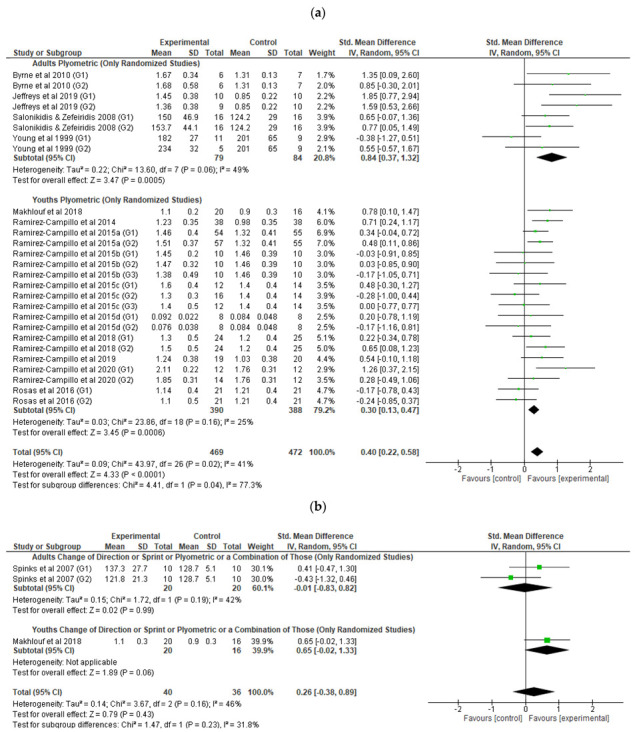
Forest plot of the overall standardized mean difference [95% CI] for each randomized study (the size of the green dot corresponds to the weight of the study within the meta-analysis) with control group with (**a**) a plyometric training intervention, and (**b**) combination of change of direction, sprint, or plyometric training intervention.

**Figure 3 healthcare-10-00593-f003:**
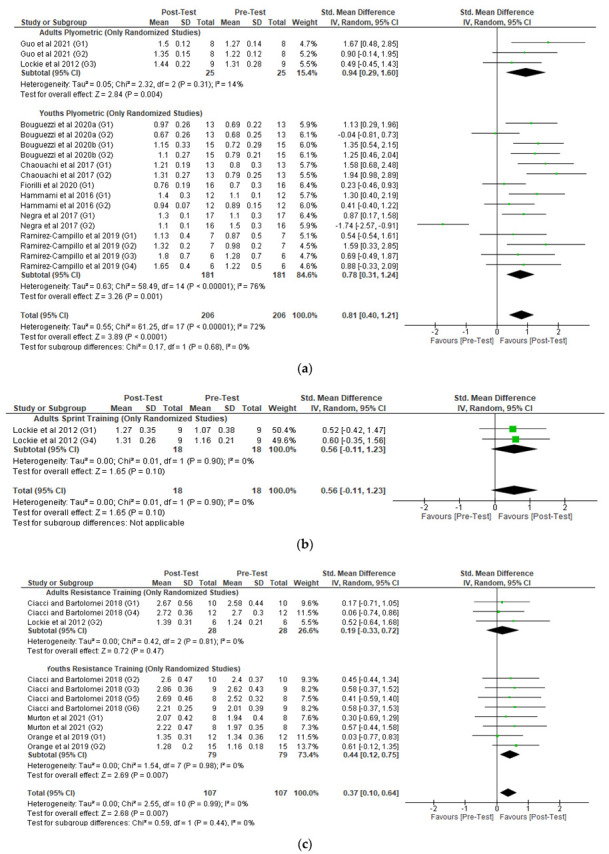
Forest plot of the overall standardized mean difference [95% CI] for each randomized study (the size of the green dot corresponds to the weight of the study within the meta-analysis), without a control group, with a (**a**) plyometric training intervention, (**b**) sprint training intervention, and (**c**) resistance training intervention included in the meta-analysis.

**Figure 4 healthcare-10-00593-f004:**
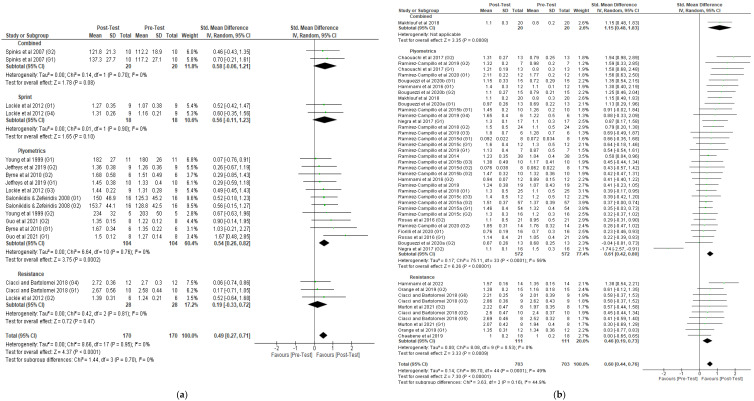
Forest plot, ranked by treatment effectiveness (the size of the green dot corresponds to the weight of the study within the meta-analysis), regarding (**a**) adult and (**b**) youth athletes.

**Table 1 healthcare-10-00593-t001:** Definition of types of interventions and comparators.

Type	Definition
Intervention	
Plyometrics	Exercises that are designed to enhance neuromuscular performance on the lower limbs. This involves application of jump, hopping, and bounding training.
Resistance training	Training program that aims to improve strength, power, or hypertrophy with resistances (e.g., elastic bands, barbells, dumbbells, kettlebells, or body weight).
Sprint training	Acceleration or maximal velocity training either resisted or unloaded.
Change of direction (COD) or sprint or plyometric or a combination of those	COD: Any exercise that enforces the participant to accelerate, decelerate and do a COD. This type of intervention is defined by a combination of one or more of sprint training, COD training, or plyometric training.
Sports-specific training	Sports-specific exercises training (e.g., small-sided games in soccer).
Control	
Maintained training routines	Sport training routines

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
