# Peer review of "How to Improve the Reactive Strength Index among Male Athletes? A Systematic Review with Meta-Analysis"

_healthcare, 2022, doi:10.3390/healthcare10040593_

Round 1
Reviewer 1 Report
In this systematic review and meta-analysis, the authors compared the effects of different training protocols on reactive strength index, finding that plyometric training has a statistically significant effect both in adults and young athletes; moreover, the authors emphasized the positive effect that resistance training has on young athletes compared to adults.
The manuscript is well written, and no major revisions are required.
Abstract
Methods are not present. There isn’t any mention on the databases used for the search, and the adherence to the PRISMA guidelines should be mentioned.
Line 12: The authors could add a sentence to clarify the reactive strength index to the reader.
Introduction
Line 62-70: Why do authors write about fatigue? It would be more accurate to report some examples of how coaches use the RSI in practice, as you mentioned before in lines 60-62.
Methods
Line 82-84: please add if you followed the PRISMA statement
Line 98-105: Inclusion and exclusion criteria should be more clear
Author Response
General Comment:
In this systematic review and meta-analysis, the authors compared the effects of different training protocols on reactive strength index, finding that plyometric training has a statistically significant effect both in adults and young athletes; moreover, the authors emphasized the positive effect that resistance training has on young athletes compared to adults.
The manuscript is well written, and no major revisions are required.
Authors:
Thank you for your kind words and for agreeing to review the article.
Comment 1:
(Abstract)
Methods are not present. There isn’t any mention on the databases used for the search, and the adherence to the PRISMA guidelines should be mentioned.
Authors:
We appreciate and agree with this comment. PRISMA guidelines and databases used were added to the abstract. Nevertheless, doing this, the number of words allowed for the abstract was overtaken. Please refer to page 1 (lines 19-22): “Electronic searches were conducted in MEDLINE, PubMed, Scopus, SPORTDiscus and Web of Science from database inception to 11 February 2022. This meta-analysis was conducted in accordance with the recommendations of the Preferred Reporting Items for Systematic Reviews and Meta-Analyses (PRISMA).”
Comment 2:
(Abstract)
Line 12: The authors could add a sentence to clarify the reactive strength index to the reader.
Authors:
Thank you for the suggestions. Changes were made accordingly. Please refer to page 1 (now lines 16-18): The reactive strength index (RSI) describes the individual’s capability to quickly change from an eccentric muscular contraction to a concentric one and can be used to monitor, assess, and reduce the risk of athlete’s injury.
Comment 3:
(Introduction)
Line 62-70: Why do authors write about fatigue? It would be more accurate to report some examples of how coaches use the RSI in practice, as you mentioned before in lines 60-62.
Authors:
Thank you for pointing this out. Fatigue portion of the text was replaced. Please refer to page 2 (now line 77-82): ”Furthermore, both RSI and RSImod can be used as variables to potentially monitor athlete’s neuromuscular readiness [22]. Moreover, the RSI has been shown to have a strong relationship with change of direction speed, acceleration speed[23], and agility [24]. Also, maximal strength, especially relative to body mass, appears to have a very strong relationship with RSImod, indicating that stronger athletes tend to have better reactive strength [25].”
Comment 4:
(Methods)
Line 82-84: please add if you followed the PRISMA statement.
Authors:
Thank you for this comment. Changes were made accordingly, please refer to page 2 (now line 90-92): This meta-analysis was conducted in accordance with the recommendations of the Preferred Reporting Items for Systematic Reviews and Meta-Analyses (PRISMA) [29]. The study protocol was registered with PROSPERO (CRD42020176616).
Comment 5:
(Methods)
Line 98-105: Inclusion and exclusion criteria should be more clear.
Authors:
Thank you for the suggestion. Please refer to page 3 (now lines 181-191):
2.4. Inclusion criteria
Scientific peer-reviewed published papers written in English, Portuguese, French, and Spanish were eligible for the present systematic review. The review sought to identify all studies reporting exercise interventions to improve the RSI in both male adult and youth athletes. Therefore, studies were eligible if: (1) subjects were male athletes; (2) subjects had between 11 and 45 years old; (3) the study included at least, two moments of evaluation with a baseline RSI measurement and a post-intervention RSI measurement; (4) the study included a training program that aimed to improve the RSI.
2.5. Exclusion criteria
Studies that do not describe a protocol to induce effects on RSI or studies that used RSI during a recovery program were excluded from the present study.

Reviewer 2 Report
healthcare-1640679_review
Title: How to Improve the Reactive Strength Index Among Male Athletes? A Systematic Review with Meta-Analysis
Comments and Suggestions for Authors
Dear authors,
I was glad to have the opportunity to review this interesting manuscript that analyzes the scientific scientific evidence about the effectiveness of different training programs on reactive strength index in male athletes (adult and youth). You also, completed this review by a qualitative synthesis (risk of bias assessment, study characteristics), and a quantitative synthesis (meta-analysis) of the selected articles.
As has been well described in the article, your results highlight the importance and need to future research with greater methodological quality, studies that include sprint interventions and combined training interventions, and studies that analyse other sport modalities. Therefore I agree with you on the importance of continuing to carry out research in this field because this information is essential to carry out better training intervention programs to improve the reactive strength index.
In general, the manuscript is well-written, the text is understandable and organized and it is easy to follow authors’ thoughts and reasoning.
In my opinion, the introduction and discussion sections are well-described.
I would like to comment on some minor issues that could be addressed to improve the document, in my opinion.
Specific comments:
Introduction
- Introduction section is adequate and complete.
Materials and methods
- Page 2, lines 81-84. What is the time period in this study was carried out? One year, six month…..You point out that until 11 February of 2022, but when did it start?
- Pages 3, lines 97-103. About the Eligibility Criteria, Why did you exclude the studies in female athletes?
Results
- Pages 4-11:
I would like to congratulate you for the work done in analysing and synthesizing all the information you have consulted. You have carried out a very detailed analysis of the information shown in the results section and in the documentation available in the supplementary file(s).
- Web appendix 2, 3 and 4: I suggest that you consider adding in the tables the corresponding numbering of the references of each of the articles (as cited in the text and in the reference list). This would make it easier to find the articles shown in the tables in the reference list.
-Your discussion section well-structured and comprehensive.
Conclusions
-Your conclusions are appropriate.
I hope that my comments could help to improve the paper. Congratulation for the original research
Author Response
General Comment:
was glad to have the opportunity to review this interesting manuscript that analyzes the scientific scientific evidence about the effectiveness of different training programs on reactive strength index in male athletes (adult and youth). You also, completed this review by a qualitative synthesis (risk of bias assessment, study characteristics), and a quantitative synthesis (meta-analysis) of the selected articles.
As has been well described in the article, your results highlight the importance and need to future research with greater methodological quality, studies that include sprint interventions and combined training interventions, and studies that analyse other sport modalities. Therefore I agree with you on the importance of continuing to carry out research in this field because this information is essential to carry out better training intervention programs to improve the reactive strength index.
In general, the manuscript is well-written, the text is understandable and organized and it is easy to follow authors’ thoughts and reasoning.
In my opinion, the introduction and discussion sections are well-described.
I would like to comment on some minor issues that could be addressed to improve the document, in my opinion.
Authors:
Thank you.
Comment 1:
(Introduction)
Introduction section is adequate and complete.
Authors:
Thank you for pointing this out. Changes were made accordingly. Please refer to page 2 (now lines 94-96): The literature search on five electronic databases (i.e., MEDLINE, PubMed, Scopus, SPORTDiscus, and Web of Science) started on 5 July 2020 and was conducted from data-base inception to 11 February 2022.
Comment 3:
(Materials and methods)
Pages 3, lines 97-103. About the Eligibility Criteria, Why did you exclude the studies in female athletes?
Authors:
Thank you for giving us the opportunity to clarify this. Although plyometric or strength training programs might also improve the RSI levels on female athletes, the improvement magnitude might be different due biological consideration (ex: lower testosterone levels and lower muscle volumes). As that, and due to heterogeneity issues, it was decided to not include female athletes on the meta-analysis. Authors recognized that it would be possible to design a meta-analysis just for the female athlete, however, this would add more figures, more tables, and more information for an already dense manuscript. Nevertheless, this is a gap in the literature that needs to be addressed on future studies.
Comment 4:
(Results)
Pages 4-11:
I would like to congratulate you for the work done in analysing and synthesizing all the information you have consulted. You have carried out a very detailed analysis of the information shown in the results section and in the documentation available in the supplementary file(s).
Authors:
We appreciate your comment.
Comment 5:
(Results)
Web appendix 2, 3 and 4: I suggest that you consider adding in the tables the corresponding numbering of the references of each of the articles (as cited in the text and in the reference list). This would make it easier to find the articles shown in the tables in the reference list.
Authors:
Thank you for pointing this out. Changes were made accordingly.
Comment 6:
(Discussion)
Your discussion section well-structured and comprehensive.
Authors: Thank you.
Comment 7:
(Conclusion)
Your conclusions are appropriate.
I hope that my comments could help to improve the paper. Congratulation for the original research
Authors: Thank you for your kind words.
